

# Fluvial organic carbon fluxes from oil palm plantations on tropical peatland

**Sarah Cook[1, 2*], Mick J. Whelan[2], Chris D. Evans[3], Vincent Gauci[1], Mike Peacock[1,4],**

**Mark H. Garnett[5], Lip Khoon Kho[6], Yit Arn Teh[7], Susan E. Page[2]**

[1] Faculty of STEM, School of Environment Earth and Ecosystems, The Open University, Milton Keynes, MK7 6AA, UK

[2] Centre for Landscape & Climate Research, School of Geography, Geology and the Environment, University of
Leicester, LE1 7RH, UK

[3] Environment Centre Wales, Centre for Ecology and Hydrology, Bangor, LL57 2UW, UK

[4] Department of Aquatic Sciences and Assessment, Swedish University of Agricultural Sciences, 750 07, Uppsala, Sweden

[5] Natural Environment Research Council Radiocarbon Facility, Rankine Avenue, Scottish Enterprise
Technology Park, East Kilbride, G75 0QF, UK

[6] Tropical Peat Research Institute, Biological Research Division, Malaysian Palm Oil Board, Bandar Baru Bangi 43000, Kajang, Selangor, Malaysia

[7] Institute of Biological and Environmental Sciences, University of Aberdeen, Aberdeen AB24 3UU, UK

*Correspondence to*: Sarah Cook (sc606@le.ac.uk)



**Abstract**

Intact tropical peatlands are dense, long-term stores of carbon. However, the future security of these ecosystems is at risk from land conversion and extensive peatland drainage. This can enhance peat oxidation and convert long-term carbon sinks into significant carbon sources. In Southeast Asia, the largest land use on peatland is for

oil palm plantation agriculture. Here, we present the first annual estimate of exported fluvial organic carbon in the drainage waters of four peatland oil palm plantation areas in Sarawak, Malaysia. Total organic carbon (TOC) fluxes from the plantation second and third order drains were dominated by dissolved organic carbon (DOC) and ranged from $34.4 \pm 9.7$ C m$^{-2}$ yr$^{-1}$ to $57.7 \pm 16.3$ g C m$^{-2}$ yr$^{-1}$ ($\pm$ 95% confidence interval). The magnitude of the flux was found to be influenced by water table depth, with higher TOC fluxes observed from

more deeply drained sites. Radiocarbon dating on the DOC component indicated the presence of old (pre-1950s) carbon in all samples collected, with DOC at the most deeply drained site having a mean age of 735 years. Overall, our estimates suggest fluvial TOC contributes ~5% of total carbon losses from oil palm plantations on peat. Maintenance of high and stable water tables in oil palm plantations appears to be key to minimising TOC losses. This reinforces the importance of considering all carbon loss pathways, rather than just $CO_2$

emissions from the peat surface, in studies of tropical peatland land conversion.



## 1. Introduction

Tropical peat carbon stocks are estimated to be 105 Gt C ($105 \times 10^{15}$ g; Dargie et al., 2017), with over half (57 Gt C) stored within the peatlands of Southeast Asia (Page et al., 2011a; Dargie et al., 2017). Thus, Southeast Asian peatlands and tropical peatlands as a whole contain approximately 10% and 20% of the global peat carbon

stocks respectively (Page et al., 2011a; Dargie et al., 2017). Disturbance, including burning, deforestation and drainage, often associated with land-use change, is common across the peatlands of Southeast Asia, driven by strong social and economic pressures to expand agricultural, palm oil and pulpwood production to support growing populations and economic development (Miettinen et al., 2012a). Consequently, only 6% of remaining peat swamp forest areas are considered pristine (Miettinen et al., 2016), whilst carbon emissions from peatlands

converted to agriculture are globally significant and increasing (Wijedasa et al., 2017).

Oil palm has played a central role in land-use change within Indonesia and Malaysia over the last few decades, driven by global consumer demand for vegetable oil-based products, and the exceptionally high productivity of oil palm compared to other oil-producing crops (Wicke et al., 2008; 2011; Schrier-Uijl et al., 2013; Gandaseca

et al., 2014; Cole et al., 2015; Wijedasa et al., 2017). Over the next 30 years around 50% of the remaining peat swamp forest in Indonesia is at risk of land conversion, predominately for oil palm cultivation, despite a recent moratorium on the issuing of new concession licences for agriculture or logging in peatlands (Wijedasa et al., 2018). Peatland oil palm expansion is also prevalent within the Malaysian state of Sarawak (SarVision, 2011; Cole et al., 2015). By early 2016, nearly half (46%) of the total peatland area in Sarawak was under industrial

plantations, with 96% of this area used for the cultivation of oil palm (Wetlands International, 2016).

The conversion of peat swamp forest to oil palm plantation involves a sequence of major disturbances, principally in the form of deforestation and drainage to optimise soil moisture conditions for cultivation (Hooijer et al., 2010; Page et al., 2011b; Schrier-Uijl et al., 2013). Prior to planting, peat surfaces are typically compacted

using caterpillar-tracked vehicles, in order to improve the rooting stability of the palms and to help with subsequent machinery movement during harvesting (Melling & Henson, 2011). These processes alter the peat's natural hydrological and biogeochemical functions, resulting in increased peat decomposition, loss of water storage and long-term subsidence (Hooijer et al., 2010). This can give rise to oxidation of soil organic matter accumulated over millennia and to significant greenhouse gas (GHG) emissions (Couwenberg et al, 2010;

Hirano et al, 2012). The result is often a reversal of the peatland carbon balance; from a net sink for atmospheric



carbon to a net source (Miettinen et al., 2017). Managed land-use types now contribute to approximately 78 % of Southeast Asia's total GHG emissions related to peat oxidation (146 Mt C yr$^{-1}$; Miettinen et al., 2017).

Previous research on the effects of peat swamp forest disturbance has predominately focused on direct

atmospheric GHG emissions from the peat surface (Couwenberg et al, 2010; Hooijer et al, 2010; Page et al., 2011; Hirano et al, 2012; Matysek et al., 2017). Until fairly recently, fluvial carbon losses received less attention, but more recent data suggest that this flux can represent a substantial fraction of the tropical peatland carbon balance (Moore et al., 2013; Evans et al., 2014; Rixen et al., 2016; Yupi et al., 2017). Fluvial total organic carbon (TOC) is typically dominated by dissolved organic carbon (DOC), with particulate organic

carbon (POC) contributing < 10% of the total flux (Moore et al., 2013; Yupi et al., 2017). Dissolved organic matter (DOM) is composed of a complex mixture of aromatic and aliphatic organic compounds, which have varying susceptibility to a range of physico-chemical and biological processes including photochemical degradation, flocculation and microbial respiration, (Cory et al., 2014; Koehler et al., 2014; Catalán et al., 2015; Logue et al., 2016; Evans et al., 2017). Over 50% of the organic carbon that is leached from tropical peat is

believed to be subsequently mineralised and emitted to the atmosphere as $CO_2$ (Wit et al., 2015). These losses represent an important potential indirect contributor to GHG emissions. The riverine transport of TOC from land to ocean also represents a significant term in the global C budget (Ciais et al., 2013) and can have substantial impacts on the biogeochemistry and ecology of coastal waters (e.g. Frigstad et al., 2013).

Previous attempts to quantify fluvial carbon losses from tropical peatlands include Moore et al. (2013); Gandois

et al. (2013); Wit et al. (2015); Rixen et al. (2016) and Yupi et al. (2016).  Moore et al. (2013) reported that losses of DOC from disturbed tropical peatlands in Indonesia were around 50% greater than those from an adjacent intact peat swamp forest.  However, this research was based on a limited number of field sites (three intact sites and five degraded sites, all of which had unregulated drainage). Additional data are needed to better understand the dynamics of DOC in more intensively managed peatland environments with controlled drainage

systems. This includes tropical peatland oil palm plantations where fluvial carbon losses remain unquantified. In addition, existing data demonstrate that the radiocarbon content of exported DOC (DO$^{14}$C) from intact tropical peat swamp forests is consistently modern (Moore et al., 2013; Gandois et al., 2014; Muller et al., 2015). DO$^{14}$C data for degraded tropical peatlands are more limited, particularly for peatland oil palm plantations. Moore et al. (2013) reported DO$^{14}$C data from five from channels in drained and deforested peatlands in Indonesia, with

mean ages of 92 to 2,260 BP, and two measurements from oil palm plantations in Peninsular Malaysia which

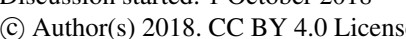



had mean ages of around 3,200 and 4,200 years BP. These limited data clearly suggest that tropical peatland

drainage releases DOC from long-term carbon stores, but are insufficient to determine whether different forms

of post drainage land-use (e.g. oil palm cultivation versus abandonment) or hydrological management (e.g.

regulated versus unregulated drainage) lead to different rates or age of DOC export.  There is a particular need

to acquire additional data from oil palm plantations as the most extensive, but currently under-represented, post-

clearance land-use on tropical peat.

In this paper, we quantify fluvial TOC concentrations, drainage channel discharge and fluvial TOC losses from

four peatland oil palm plantations in Sarawak, Malaysia, over the course of one year. The main objectives of the

study were: (i) to quantify DOC and POC transfers in channels draining peatland oil palm plantations; (ii) to

derive annual area-specific TOC flux estimates from oil palm plantations for both the wet and dry seasons and

(iii) to establish the age and quality of the DOC being lost.

## 2. Materials and Methods

### 2.1 Study site

The study was conducted in two adjacent oil palm estates: Sebungan (SE) and Sabaju (SA), situated in the

Malaysian state of Sarawak, northern Borneo, east of the town of Bintulu (between 3°07.81' N and 3°14.91'N

and 113°18.72' E and 113°32.19'E). The climate in this region is characterised by high temperatures (around

26°C) throughout the year. Annual precipitation is typically between 3000 and 3200 mm (Environmental Impact

Assessment, 2006). The annual rainfall pattern is influenced by the Northeastern (October-January) and the

Southwestern (May-August) Monsoons. The former is responsible for just under half of the annual rainfall,

making this period the wettest, while the latter contributes around a quarter.

The estates are managed by the Sarawak Oil Palms Berhad (SOPB) and cover a total area of 9,614 ha. The

Sebungan estate is established on an elliptical peat dome (Fig 1.) formed between two rivers (Batang Kemena

and Sungai Sebungan) which provide the main regional drainage for the plantation. The maximum peat depth is

5.6 m (Environmental Impact Assessment, 2006). Oil palm planting began in 2007 (making it the oldest

plantation in this study) and the total planted area is 1,648 ha.  Due to the age of the plantation, the semi-mature

palms provide a partially closed canopy (70% closed) and some shade for the peat surface.





The Sabaju estate is divided into four individual oil palm plantation s (Sabaju 1-4; Fig 1.). These are located on an irregularly shaped peat dome, with some small mineral soil hills protruding above the peat surface. Peat coring throughout the Sabaju estate has revealed very deep peat (8 m+) within Sabaju 4, moderately deep peat in Sabaju 3 (~4 m) and shallower peat in Sabaju 1 (~2 m). Sabaju 2 was excluded from this study due to the

dominance of mineral soil. Of the three oil palm plantations under investigation (Sabaju 1, 3 and 4), the youngest palms were in Sabaju 4 (1,638 ha) which was planted in 2011, followed by Sabaju 3 (1,714 ha) planted in 2010. The oldest palms are located in Sabaju 1 (2,526 ha) planted in 2008.

Plantation management on both estates is typical of other peatland oil palm plantations in this region. Artificial

drainage networks have been established to lower water tables (with a target range of -60 cm to -40 cm below the peat surface to optimise palm growth). The drainage networks consist of a grid of interconnecting ditches (Fig. 2a). The edges of the planting blocks (approximately 19 - 50 ha in area) are defined by a set of roads that provide access (Fig. 2a). Each planting block contains multiple parallel "first order" field drains (ditches approximately 0.5m deep and 1m wide) at a spacing of every 4 planting rows which feed into a central "second

order" collection drain (Fig. 2b). Second order drains subsequently feed into a system of main "third order" drains, which run parallel to the edge of the planting blocks. Water from the third order drains feeds into perimeter (ring) drains, which discharge into the adjacent river network (Fig. 2b). The hydrology of these agricultural landscapes is intensively managed throughout the year. Channel gradients are low, which means that water depths and flow directions can be controlled via channel alterations (e.g. using sand bags and boards

to narrow channels and or obstruct water flow) and the installation of weirs. This allows some mitigation for the effects of extreme environmental conditions e.g. drought and flooding.

### 2.2 *Water sample collection*

A mixture of third order (main) and second order (collection) drains were monitored over an approximately one-

year (54 week) period, from 3rd August 2015 to 8th August 2016, in the four main plantation study areas; Sebungan, Sabaju 1, Sabaju 3 and Sabaju 4. Sampling frequency was typically every 1-3 weeks. At each sampling, two water samples (1 x DOC; 1 x POC) were collected from each of the monitored channels. Samples for DOC determination were collected in pre-rinsed 60 ml Nalgene® wide-neck bottles. Water samples for POC determination were collected using 500 ml plastic bottles. Water temperature (°C), pH and electrical

conductivity (EC; μS cm$^{-1}$) were recorded in the unfiltered water samples *in situ*, using a portable pH and EC probe (Hanna HI 9813-6).



DOC samples were filtered in a field laboratory through 0.45 μm cellulose nitrate membrane filters which were

pre-rinsed with sample, using a hand-held vacuum pump, within 24 hours of collection. After filtration, water

samples were stored in the dark at 4°C (for approximately 12 weeks), to ensure DOC preservation (Cook et al.,

2016), before being shipped back to the UK. On return to the UK samples were analysed on a Shimadzu Total

Organic Carbon analyser as non-purgeable organic carbon (NPOC), to generate measured DOC concentrations.

See Supplementary Information (Text S1) for further details.

POC concentrations were determined gravimetrically. Briefly, samples were filtered through 0.4 μm Advantec

glass fibre filters which were dried for 3 hours at 105°C, weighed and combusted in a furnace for a further three

hours at 550°C and reweighed. The particulate organic matter (POM) concentration was calculated from the

difference in the filter mass between oven drying and combustion divided by the volume filtered. This was

subsequently converted into POC assuming a 50% organic carbon content (Hope et al., 1994; Moore et al.,

2011).

*2.3 Carbon quality*

The nature of the DOC sampled was investigated using *SUVA*$_{254}$ (Specific Ultra-Violet Absorption).  This is the

ultra-violet (UV) absorbance at 254 nm normalised to sample DOC concentration (Weishaar et al., 2003).

Absorbance at 254 nm is commonly used as a surrogate for DOC aromaticity, i.e. the fraction of DOC

comprised of aromatic humic substances, which absorb light in this particular part of the electromagnetic

spectrum. High-*SUVA*$_{254}$ compounds tend to be more photodegradable and low- *SUVA*$_{254}$ compounds more

biodegradable (Jones et al., 2016).

UV-vis absorbance was measured using a Cole-Parmer UV/visible spectrophotometer (230 VAC, 50 Hz) at 254

nm.  Samples were analysed immediately after filtration. *SUVA*$_{254}$ values (l mg-C$^{-1}$ m$^{-1}$) were calculated from

$$SUVA_{254}=100.\frac{A_{254}}{c_{DOC}} \qquad\qquad (1)$$

where $A_{254}$ is absorbance at 254nm and $C_{DOC}$ is the DOC concentration (mg C l$^{-1}$), after Weishaar et al. (2003).

Water samples with very high absorbance at 254nm (> 3.0) saturated the spectrophotometer and were therefore

removed from the data set (*n=1*) prior to analysis.





*2.4 Hydrology*

Drainage channel discharge was determined periodically at a number of locations under different conditions
using dilution gauging (Hongve, 1987; Hudson and Fraser, 2002). Briefly, a sodium chloride tracer solution was
injected into the flow and the consequent concentration change was observed at a point downstream ($\sim$ 30 m;
assuming full mixing has occurred). Electrical conductivity was used as a surrogate for concentration via site-
specific concentration-conductivity calibrations. This technique is considered by us to be superior to methods
based on cross sectional area and velocity measurements in shallow, irregular channels. The measured
discharge data were used to construct rating curves (stage-discharge relationships) for each location, where stage
height was measured from stage boards which were calibrated against semi-continuous recordings of water level
measured at 1-hour intervals using atmosphere-corrected pressure transducers (Mini-Divers, Schlumberger,
D1501) installed in stilling wells. Whenever water samples were collected, stage was noted and converted to
discharge using the rating curve. The rating curve equations for all measured sites, along with the standard error
of the estimate (SEE) derived from the regression equations, are presented in the Supplementary Information
(Table S1; Figs. S1 to S3). The stage records from the Mini-Divers were used to reconstruct a continuous record
of discharge for eight stations throughout the year from 15[th] September 2015 until 31[st] August 2016.

Peatland water table depths (below the peat surface) were determined using dip wells constructed from 32 mm
diameter PVC tubes cut to 2 m lengths. 5 mm holes were drilled down each tube, spaced at 35 mm intervals. The
top of each tube was fitted with a removable cap to allow access to the tube but preventing rain and debris from
entering between measurements. The bottom of each tube was fitted with a glued PVC plug to prevent sediment
encroachment. A cluster of three dip wells (inserted 1.5 m into the peat from the surface, at 0.5 m intervals from
one another) was installed in the centre of one planting block at each plantation study site. Measurements of water
table depths were made using a dipmeter (*In-situ* Rugged Water Level TAPEs) in each field sampling visit, at the
same time as water samples were collected.

*2.5 Flux calculation*

The annual area-specific TOC flux ($J$: g C m$^{-2}$ yr$^{-1}$) for each station was calculated from

$$J = C_W.R_E = \frac{\Sigma(C_i Q_i)}{\Sigma Q_i} . R_E \qquad (2)$$





where $C_W$ (g C m$^{-3}$) is the annual flow-weighted concentration, $C_i$ (g C m$^{-3}$) is the instantaneous (sampled)

concentration of TOC (DOC+POC) on sampling date $i$, $Q_i$ is the corresponding discharge at time of sampling

(m$^3$ s$^{-1}$) and $R_E$ is the annual runoff (m yr$^{-1}$).

$R_E$ can be calculated from measured channel discharge and catchment area:

$$R_E = \frac{\sum Q_A}{A} = \frac{\sum_h^T Q_h}{A}$$

(3)

where $Q_A$ is the measured mean annual discharge (m$^3$ yr$^{-1}$) estimated as the sum of hourly discharge $h$ ($Q_h$; m$^3$ h$^{-1}$)

$^1$) over $T$ hours in the measurement period and $A$ is the catchment area (m$^2$).

In principle, the catchment area for a particular drain can be estimated from the topography of plantation block

areas which it serves.  However, because channel gradients in peatland landscapes are low, this is subject to a

high level of uncertainty, particularly for lower order drains. Consequently, we applied a water balance

approach to the calculation of discharge. To do this, $R_E$ was assumed to be the same for all plantation sites,

based on the assumption that all sites were hydrologically similar in terms of the annual water balance: all sites

had similar soil properties, topography, vegetation and management and were sufficiently close together such

that they experienced very similar rainfall. by also assuming no annual change in catchment water storage (the

average difference between water table depth at the beginning and end of the monitoring period was ~ 30 mm,

implying that any storage changes were in the order of 10s of mm), $R_E$ can be calculated from climate data using

$$R_E = P \text{ - } ET_a$$

(4)L

where $P$ is the annual rainfall (mm yr$^{-1}$) and $ET_a$ is the annual actual evapotranspiration (mm yr$^{-1}$) which was

calculated from

$$ET_a = k_C . ET_O$$

(5)

where $k_C$ is the so-called "crop coefficient" and $ET_0$ is the annual reference evapotranspiration rate (a

standardised ET rate which assumes soil moisture is not limiting).

$ET_0$ (mm yr$^{-1}$) was calculated from average daily values of temperature, relative humidity, wind speed and net

radiation flux density using the Penman Monteith equation (e.g. Monteith, 1965). Meteorological data





(including $P$) were available over the period 15$^{th}$ September 2015 to 6$^{th}$ August 2016 from an on-site automatic

weather station (Davis Vantage Pro 2). Carr (2011) reports that $k_C$ for oil palms typically varies between 0.8

and 1 when soil moisture is unlimited in the wet season. Lower $k_C$ values have been reported in the dry season

for other areas. However, although the near surface peat layer does dry out seasonally, palm oil roots generally

5    extend much deeper (i.e. 50-70 cm; Othman et al. 2010; Veloo et al. 2015) – into or close to the saturated zone.

We therefore assumed that soil moisture availability is rarely limiting, and adopted a value of 0.9 for $k_C$ over the

whole year. This assumption was supported by water balance modelling following the method of Whelan &

Gandolfi (2002), which showed that measured discharge at all Sebungan monitoring stations could be simulated

well with $ET_a \approx 0.9\ ET_0$ over the whole year (data not shown).

*2.6 Flux uncertainty*

TOC fluxes were subject to considerable uncertainties. Specifically, in (i) DOC concentration; (ii) the mean

annual runoff ($R_E$) and (iii) channel discharge at the time of sampling ($Q_i$). These were accounted for in our

overall flux estimates using a Monte Carlo Simulation approach as detailed below.

*2.6.1 Monte Carlo Simulation*

The second order drain with the most reliable catchment area was SE 1 (2,167,300 m$^2$). This was defined

topographically during a field reconnaissance in April 2015 in which water flow directions and the drainage

layout were mapped manually and then digitised in ArcGIS. A value of $R_E$ was then calculated by applying

20    Equation (3) for the period for which complete meteorological data was available (15$^{th}$ September 2015 to 6$^{th}$

August 2016) and for the whole year (discharge measurements were available for the whole year). Their

associated error was then estimated using standard errors derived using a Monte Carlo error propagation

simulation (e.g. Iman and Conover, 1980; Farrar et al., 1989) in which the error in $Q_h$ ($\pm$ 15.14 m$^3$ h$^{-1}$) was

assumed to be the standard error of the estimate in the rating curve for SE 1 and in which an error of 25% was

25    arbitrarily (and conservatively) assumed for $A$. Briefly, values for each variable were selected randomly from

probability density functions (pdfs) and employed in Equation (3) in a large number (5000) of iterations.

Gaussian pdfs were used based on the assumption that the estimate of a statistic is normally distributed about the

true value (central limit theorem) with the best-estimate value assumed for the mean and SEE assumed for the

standard deviation. This is described in more detail in the Supplementary Information (Text S2; Fig.S4 to S7).



The following variables were sampled from their pdfs in calculating flux uncertainty in Equation (2) for each

plantation area: $C_i$, $Q_i$ and $R_E$. Variances for $Q_i$ and $C_i$ were derived, respectively, from (i) the SEE values given

in the regression equations for the rating curves and (ii) the error associated with the DOC concentrations

obtained using the TOC analyser (assumed to be the true value) with a precision of ~5% (Graneli et al. 1996;

Bjorkvald et al. 2008; Shafer et al. 2010). This is detailed in the Supplementary Information (Text S1).

*2.7 Radiocarbon dating (DO$^{14}$C)*

Water samples for radiocarbon dating were acquired from three second order drains within both the Sebungan

and Sabaju 3 plantations (6 samples in total). Sample collection from third order (main) drains was avoided to

prevent pseudo-replication (i.e. nested catchments). All samples were collected over the course of 24 hours in

the wet season (April 2016) in pre-rinsed (with sample) 500 ml polypropylene bottles and filtered using a 0.7

μm glass fibre filter. These filters were pre-combusted at high temperatures (450°C) to minimise the organic

matter contamination risk. The water samples were stored at 4°C for approximately one year prior to analysis.

Cold storage has been shown to be a viable method for the long-term preservation of carbon isotopic signatures

(Gulliver et al., 2010). Samples were analysed by accelerator mass spectrometry (AMS) at the Natural

Environment Research Council facility in East Kilbride, UK in 2017. Values were expressed as %modern (*m*)

or conventional radiocarbon ages (in years BP, where 0 BP = AD 1950 = 100 %modern):

$$Age \text{ (years BP)} = -8033. \text{Ln } (m/100) \tag{6}$$

$$m = \left(\frac{A_{SAMPLE}}{A_{OXALIC}}\right).100 \tag{7}$$

where $A_{SAMPLE}$ is the $^{13}$C-normalised radioactivity in the sample and $A_{OXALIC}$ is the $^{13}$C-normalised radioactivity

in the oxalic acid international radiocarbon standard with a radioactivity equivalent to the atmosphere in 1950

(i.e. 100 %modern = year AD 1950).

*2.8 Age attribution model*

DOC in water samples contains a mixture of organic matter from old ($^{14}$C depleted) peat and recently

photosynthesised ($^{14}$C enriched) litter and plant material (Evans et al., 2014; Campeau et al., 2017). Ascribing a

'mean age' to carbon fixed post-1950 is complicated by nuclear bomb testing which released a pulse of enriched





[14]C into the atmosphere. The [14]C isotopic signature is, therefore, likely to reflect a mixture of both old (pre-

1950's) and new (post bomb/[14]C enriched) carbon, in varying amounts. Thus, no single definitive 'mean age'

can be ascribed to the sample (Evans et al., 2007). To address this, the age attribution model previously

described by Moore et al. (2013) and Evans et al. (2014) was used to infer an indicative age distribution for the

DOC in the samples. The model assumes an exponential decrease in the amount of DOC produced with

increasing depth (and therefore age) within the peat profile. Each year class in the profile was assigned a [14]C

value based on estimated atmospheric $CO_2$ for that year (see Evans et al., 2014) and the maximum age was set

to 4,300 years BP, based on [14]C basal ages recorded for this region by Dommain et al. (2011). The model can be

expressed as:

$$DO^{14}C = \sum_{t=1}^{t=4,300} ({}^{14}CO_{2t} \cdot e^{(-k,t)}) \tag{8}$$

where $DO^{14}C$ is the measured [14]C of the DOC sample, $t$ is year prior to present day, $^{14}CO_{2t}$ is the [14]C level of

atmospheric $CO_2$ in year $t$ and $k$ is an exponential decay constant with a value between 0 and 1. For each sample,

$k$ was adjusted to fit measured $DO^{14}C$ using an iterative optimisation routine (Microsoft Excel Goal Seek).

Modelled age distributions were summarised by aggregating year classes into the age categories 0-9, 10-49, 50-

99, 100-299, 300-499, 500-699, 700-999, 1000-2999 and 3000+ years.

*2.9 Bulk density and aerated carbon stocks*

Bulk density (BD) was determined on four peat cores (up to 4 m in length) per plantation. Subsamples of 123

cm³ volume were taken every 15 cm down to the water table (identified using a dipmeter) and every 50 cm

thereafter.  A total of 47 peat samples were oven dried at 105°C for up to 120 hours (until the dry weight of the

sample had stabilised) and weighed to calculate the BD. The total aerated carbon stock $SOC_{air}$ (kg m$^{-2}$) for each

plantation was then derived (Tiemeyer et al., 2016) as:

$$SOC_{air} = \frac{z_{WT}}{n} \cdot \sum \rho_j \cdot SOC_j \cdot \tag{9}$$

where $\rho_j$ (g cm³) is bulk density for sample $j$, $SOC_j$ is the soil organic carbon content (g kg$^{-1}$) of sample $j$

(derived from loss on ignition assuming 50% carbon), $n$ is the number of samples collected above the water

table and $z_{WT}$ is the average annual water table depth (m).



*2.10 Statistical analysis*

Statistical analysis was performed using GraphPad Prism v7. The threshold level of statistical significance was set at a probability of 0.05 but greater significance was also noted. For multiple comparisons, one-way ANOVAs were performed. The assumptions that the data adhered to normality and homogeneity of sample variance were

checked *a priori* using the Shapiro-Wilk and Bartlett tests, respectively. If significant differences between the group means were identified then a *post-hoc* test was carried out. In cases where these assumptions were not valid, the non-parametric Kruskal-Wallis test was performed together with a *post-hoc* test as above. In addition, the relationship between variables was tested using linear regression models.

**3. Results**

*3.1 Water table depths and aerated carbon stocks*

Mean water table depths for the individual estates are displayed in Fig. 3 and Table 1. The lowest mean water table (i.e. furthest from the peat surface) was observed in the Sebungan estate (-55 cm; Fig. 3a). The highest mean water table (i.e. closest to the peat surface) was observed in Sabaju 1 (-31 cm; Fig 3a). The SE site displayed the greatest degree of water table variability, with depths ranging from -128 cm to + 5 cm (i.e. above

the peat surface). This was closely followed by Sabaju 3, where water depths ranged from -82 cm to + 4 cm. The lowest variability was observed at Sabaju 1 where the water table depth varied between -53 cm and -6 cm. Seasonal variations in water table followed monthly rainfall (Fig. 3b).

BD measurements revealed higher compaction of the near-surface peat within the Sebungan estate compared to

that in the Sabaju estate; the average BD for the upper 1 m of peat in the Sebungan estate was $0.18 \pm 0.01$ (standard error) g cm$^{-3}$, compared to an average BD value for the Sabaju estate of $0.10 \pm 0.01$ g cm$^{-3}$ (Table 1). Overall, BD ranged from $0.06 \pm 0.01$ g cm$^{-3}$ to $0.24 \pm 0.01$ g cm$^{-3}$ across all samples. Similarly, the Sebungan estate displayed a higher overall $SOC_{air}$ value of 45.3 kg m$^{-2}$ which was more than double the $SOC_{air}$ values calculated for the three Sabaju estates (14.2 to 18.9 kg m$^{-2}$; Table 1).

*3.2 Catchment hydrology*

Totals for $P$, $ET_0$, $ET_a$ and $R_E$ over the period 15$^{th}$ September 2015 to 6$^{th}$ August 2016 were 2046 mm, 1135 mm, 1021 mm and 1025 mm respectively. The value for $P$ was much lower than the typical annual value in the region (*ca* 3000mm yr$^{-1}$). This was due to the influence of the 2015/2016 El Niño event characterised by



extended dry periods. The calculated $R_E$ was $1022 \pm 55$ mm for the period for which complete meteorological

data were available and $1090 \pm 147$ mm yr$^{-1}$ for the whole year. These values match the $R_E$ estimate derived

from the water balance (1025 mm), lending confidence to our estimates of TOC fluxes.

*3.3 Fluvial organic carbon*

Mean annual TOC concentrations ranged from 29.3 to 51.2 mg l$^{-1}$ and 29.6 to 49.6 mg l$^{-1}$ and in the second order

collection drains and third order main drains, respectively (Table 2). The Sebungan plantation sites displayed the

highest TOC concentrations and those in Sabaju 3 the lowest (Table 2). Concentrations in both the main and

collection drains at Sebungan were significantly different to all the other plantation sites ($p < 0.0001$, unpaired

Kruskal-Wallis). DOC was always the dominant component of TOC, accounting for 84 – 95 % of TOC (Table

2). Overall, there were no significant seasonal temporal trends in TOC concentrations, which remained

relatively stable for the duration of the investigation (~30 mg l$^{-1}$ to 50 mg l$^{-1}$; Fig. 4). There were no systematic

differences in $SUVA_{254}$ between the different estates, or between second and third order drains.

TOC fluxes were determined for all individual plantations except for three second order drains (one in Sabaju 1

and two in Sabaju 3), which experienced significant changes in their hydrological regimes as a result of

plantation management actions. These actions included channel blocking and widening which led to

complications in estimating discharge and therefore TOC fluxes. Mean annual TOC fluxes ranged from $34.4 \pm$

9.7 g C m$^{-2}$ yr$^{-1}$ to $57.7 \pm 16.3$ g C m$^{-2}$ yr$^{-1}$ (Fig. 5; Table 2; see Supplementary Table S2 for individual site

fluxes,), with significantly higher fluxes ($p < 0.05$, unpaired, one-way ANOVA) recorded in the Sebungan

plantation and the lowest recorded in Sabaju 3 (Fig. 5). Mean annual TOC losses were $44.7 \pm 12.6$ g C m$^{-2}$ yr$^{-1}$

and $42.1 \pm 11.9$ g C m$^{-2}$ yr$^{-1}$ from the second and third order drains, respectively. The two components of TOC

(DOC and POC) contributed 93 % and 7 %, respectively, on average to annual TOC yields across all plantation

sites, with a slightly higher contribution from DOC to TOC in the second order drains (91 % third order vs 94 %

second order).

*3.4 DO$^{14}$C*

The greatest $^{14}$C-enrichment was exhibited by site SA 3.6 (102.6 % modern) and the greatest $^{14}$C-depletion by

site SE 4 carbon (91.3 % modern; Table 3). Conventional mean $DO^{14}C$ age was positively correlated with the

depth of the water table and drainage intensity (Table 3; Fig. 6), but this was strongly dependent on site SE 4,





which had the deepest average drainage depth and greatest $^{14}C$ depletion, corresponding to a mean $DO^{14}C$ age of

$735 \pm 37$ years BP (Table 3). The other five sites were all wetter (average water tables -21 to -50 cm) and

clustered within a fairly narrow $DO^{14}C$ range (99.6 – 102.6 %modern).

The fitted age attribution model (Fig. 7) suggests that the majority of DOC in all samples, other than SE 4,

originates from peat with a $^{14}C$ age of 100 - 500 years BP. Based on its lower measured $^{14}C$ value, the SE 4

sample is estimated to contain a larger proportion of older peat carbon, with > 35% estimated to be derived from

material with a $^{14}C$ age exceeding 1000 years BP.

### 4. Discussion

Average DOC concentrations in water draining the Sabaju oil palm plantations were lower than those reported

by Moore et al. (2013) for drained tropical peatlands (*ca* 52 mg L$^{-1}$), whilse those from the Sebungan plantation

were in line with the values reported by Moore et al. (2013). The average annual TOC fluxes (Fig. 5) for the

third order (main) and second order (collection) drains were also less than TOC flux estimates for drained

tropical peat swamp forests reported elsewhere (94 to 108 g C m$^{-2}$ yr$^{-1}$; Moore et al., 2013; Muller et al., 2015)

and those reported for intact peat swamp forests in Indonesia and Sarawak, Malaysia (63 - 64 g C m$^{-2}$ yr$^{-1}$) by

Moore et al. (2013) and Muller et al. (2015). The annual Sebungan estate TOC fluxes were significantly higher

than the fluxes from the Sabaju estate (Fig. 5). The lower annual fluxes (for drained tropical peat) reported here

reflect a combination of relatively low TOC concentrations (e.g. compared to Moore et al, 2013) and relatively

low total annual runoff during the study period. The latter was influenced by an El Niño-driven drought event

(i.e. low rainfall recorded in 2016; Fig. 3b).  Since discharge varies much more than DOC concentration, the

temporal pattern of DOC fluxes from the study catchments is primarily controlled by discharge (Clark et al.,

2007). Initial plantation development on tropical peat is often associated with the release of large pulses of

carbon due to enhanced mineralisation during the first 5 years (Hooijer et al., 2010; Page et al., 2011b). Since

the plantations which were sampled here were 6 to 9 years old, these initial responses to disturbance are unlikely

to have been captured.  Together with the low (El Niño related) rainfall and runoff rates experienced, our flux

estimates are, therefore, probably relatively conservative of fluvial TOC losses from oil palm plantations

overall.  Reported peat surface $CO_2$ emissions from oil palm plantations are typically in the range of 900 g C m$^{-2}$

yr$^{-1}$ (Husnain et al., 2014) to 2700 g C m$^{-2}$ yr$^{-1}$ (Hooijer et al., 2012). As such, the TOC fluxes reported from this

study could represent an additional carbon loss equal to ~2% to 5% of total carbon emissions from oil palm

plantations on peat.



The spatial variations in the TOC flux across the four monitored oil palm plantations (Fig. 5) were principally controlled by DOC concentrations which, in turn, appeared to be related to water table depth, with higher DOC concentrations and fluxes from the deep-drained Sebungan site compared to the shallower-drained Sabaju sites

(Figs. 3 and 5; Table 2 and 3).  This is broadly consistent with the conclusions of previous analyses of peatland drainage impacts on DOC loss (Evans et al., 2016), but contrasts somewhat with the assessment by Moore et al. (2013), which recorded higher DOC fluxes (principally due to higher water losses) but not higher DOC concentrations. On the other hand, Yupi et al. (2017) did record higher DOC concentrations as well as fluxes from a small drainage-affected catchment, compared to a river draining a larger, relatively intact peat swamp

forest. Differences between our results and those of Moore et al. may be explained by differences between the study areas; the Moore et al. study compared highly contrasting sites (drained and deforested versus undrained natural forest) with large resulting differences in water balance. In contrast, our study compared sites with differing drainage depths within a single land-use category and consequently a more uniform water balance. On the other hand, we observed variations in $DO^{14}C$ versus drainage depth that were consistent with those observed

elsewhere in both tropical and high-latitude peatlands (Evans et al., 2014) (Fig. 6 and Table 3). This clearly suggests that deeper drainage leads to the mobilisation of older peat C into DOC and is also associated with the release of more humified (high $SUVA_{254}$) material, consistent with previous findings (Olefeldt et al., 2013). In general, our observed $SUVA_{254}$ values were higher than those reported previously in runoff from intact peat swamp forests (Moore et al., 2013; Gandois et al., 2013) which can be explained by a transition from plant-

derived to peat-derived DOC sources following forest clearance and drainage (Kononen et al., 2016).

The differences in watertable depth (Fig. 3) between the Sabaju and Sebungan estates are further accentuated by the differences in bulk density, leading to large differences in $SOC_{air}$ (Table 1). This suggests a denser concentration of carbon in the peat above the water table (i.e. higher $SOC_{air}$) which could promote higher rates

of organic matter decomposition and, hence, TOC production in the Sebungan estate. The differences in these bulk densities could originate from differences in the site management, plantation age or intrinsic differences in the peat characteristics between the two peat domes.

Overall, our results suggest that the clearance and drainage of peat swamp forests for oil palm plantation leads to

increased loss of carbon via fluvial pathways, in addition to recognised increases in $CO_2$ emissions. Further, our

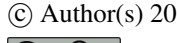



data suggest that deeper drainage within plantations leads to greater DOC export, while the greater [14]C-inferred DOC age and higher $SUVA_{254}$ indicate that this C derives from deeper within the peat profile. Together with previous studies, our results suggest that the riverine export of DOC to coastal waters from peatland regions of Southeast Asia may have increased as a result of drainage and land-conversion, with potentially profound (but

as yet uncertain) impacts on coastal marine ecosystems via altered energy and nutrient supply, pH and light regime (Durako et al., 2010; Frigstad et al., 2013; Traving et al., 2017).

With regard to the management of oil palm plantations on peatland, our measurements showed that all three Sabaju plantation sites had mean water table depths that were above the Roundtable on Sustainable Palm Oil's

(RSPO) target range of 40 to 60 cm below the surface, whereas the Sebungan site fell within this range. While the Sabaju plantation had lower DOC fluxes, and all sites had lower rates of DOC loss than the highly degraded peatland sites studies by Moore et al. (2013), the [14]C-depleted DOC measurements (relative to previous data from undrained peat swamp forests) obtained from all study sites indicate release of older stored carbon.  This suggests that even RSPO-compliant plantations may still be expected to experience elevated fluvial loss of

previously stored peat carbon, which is also indicative of drainage-induced C loss more generally (Evans et al., 2014). Managing oil palm plantations on peat to minimise both gaseous and fluvial carbon losses thus remains a significant challenge, requiring coordination between governments, the plantation industry and academia (Wijedasa et al., 2016). However, our results suggesting relationships between drainage depth and DOC concentration, flux and [14]C-inferred source indicate that any measures that enable oil palm cultivation to be

maintained at higher water levels should lead to commensurate reductions in peat carbon losses and would, therefore, lower the broader environmental impacts of oil palm cultivation.

**Acknowledgments**

This work was supported by the Natural Environment Research Council (NERC; grant: X402NE53), the Malaysian Oil Palm Board (grant: R010913000) and the AXA Research Fund. We are grateful to the University

of Aberdeen, the University of St. Andrews, Sarawak Oil Palms Berhad for additional financial support. We thank the NERC Radiocarbon Facility (2049.0317) for assisting with the water sample radiocarbon dating. We also thank L.K.K and the Tropical Peat Research Institute for field assistance and support. S.C., S.E.P., M.J.W., V.G., and C.D.E conceived, designed and implemented the study. S.C performed the Malaysian field data collection and analysed the data along with M.P. All authors discussed the results and contributed to the writing of the

manuscript. Data is available in the accompanying supporting information.





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





**Tables**

**Table 1.** Mean water table, bulk density and $SOC_{air}$ measurements for the four oil palm plantations; ± represents

the standard error of the mean.

| Plantation | Mean water table depth (cm) | Average bulk density (g cm$^{-3}$) | | $SOC_{air}$(kg m$^{-2}$) |
| --- | --- | --- | --- | --- |
| | | Whole core | Top 1m of peat | |
| Sabaju 1 | 31.3 ± 1.1 | 0.095 ± 0.008 | 0.102 ± 0.009 | 14.7 ± 1.6 |
| Sabaju 3 | 35.1 ± 1.8 | 0.081 ± 0.005 | 0.098 ± 0.007 | 14.2 ± 2.3 |
| Sabaju 4 | 40.9 ± 1.7 | 0.092 ± 0.005 | 0.103 ± 0.012 | 18.9 ± 1.2 |
| Sebungan | 55.3 ± 3.3 | 0.165 ± 0.008 | 0.182 ± 0.007 | 45.3 ± 2.1 |



**Table 2.** Fluvial organic carbon data for the monitored oil palm plantations. DOC, POC and TOC concentrations and $SUVA_{254}$ are shown as site mean ± standard error of the mean. Annual TOC fluxes are shown as site mean ± the 95% confidence interval (CI; standard error x 1.96) which encompasses the propagated error associated with uncertainty in the DOC concentration, discharge and annual runoff derived from the Monte Carlo Simulation. $SUVA_{254}$ values are means of samples collected for each drain type in each plantation throughout the sampling year.

| Plantation | Drain type | No. of channels | Mean DOC concentration (mg l⁻¹) | Mean POC concentration (mg l⁻¹) | Mean TOC concentration (mg l⁻¹) | Annual TOC flux (g C m⁻² yr⁻¹) | $SUVA_{254}$ (L mg-C⁻¹ m⁻¹) |
|---|---|---|---|---|---|---|---|
| Sabaju 1 | Collection (2ⁿᵈ order) | 2 | 31.1 ± 1.1 | 2.7 ± 0.3 | 33.8 ± 1.1 | 38.2 ± 10.8 | 5.2 ± 0.1 |
|  | Main (3ʳᵈ order) | 1 | 29.7 ± 0.9 | 3.9 ± 0.5 | 33.6 ± 0.8 | 36.8 ± 10.5 | 5.2 ± 0.1 |
| Sabaju 3 | Collection (2ⁿᵈ order) | 2 | 25.2 ± 0.8 | 4.1 ± 0.3 | 29.3 ± 0.7 | 36.4 ± 10.3 | 5.4 ± 0.1 |
|  | Main (3ʳᵈ order) | 1 | 25.0 ± 1.0 | 4.6 ± 0.5 | 29.6 ± 1.1 | 34.5 ± 9.7 | 5.1 ± 0.1 |
| Sabaju 4 | Collection (2ⁿᵈ order) | 2 | 34.1 ± 0.6 | 2.5 ± 0.3 | 36.7 ± 0.6 | 42.1 ± 11.9 | 5.0 ± 0.1 |
|  | Main (3ʳᵈ order) | 2 | 35.3 ± 0.6 | 2.7 ± 0.3 | 38.0 ± 0.7 | 43.1 ± 12.2 | 5.6 ± 0.1 |
| Sebungan | Collection (2ⁿᵈ order) | 3 | 48.2 ± 0.8 | 3.4 ± 0.4 | 51.2 ± 1.0 | 56.3 ± 15.9 | 5.6 ± 0.2 |
|  | Main (3ʳᵈ order) | 1 | 47.1 ± 0.8 | 2.5 ± 0.2 | 49.6 ± 0.8 | 53.0 ± 15.0 | 6.4 ± 0.1 |





**Table 3.** Mean and standard errors for radiocarbon $DO^{14}C$ expressed as % modern and in conventional radiocarbon years (years BP, relative to CE 1950), expressed at ± 1σ level, for individual sample sites across the Sebungan and Sabaju estates. Mean $^{14}C$ levels >100% modern cannot be assigned an age and are subsequently referred to as 'modern'. Mean annual water table depth data for each site are also presented, along with the maximum and minimum water table depths recorded and the % of time the water table was more than 60 cm from the peat surface. Negative numbers indicate distance below the peat surface.

| Plantation | Sample site | $^{14}C$ (%modern) | $DO^{14}C$ Age (years BP) | Water table depth (cm) | | | % of time watertable was below – 60 cm |
|---|---|---|---|---|---|---|---|
| | | | | mean | maximum | minimum | |
| | SE 2 | 100.10 ± 0.46 | modern | -21 | 5 | -58 | 27 % |
| Sebungan | SE 3 | 99.57 ± 0.46 | 35 ± 37 | -52 | -34 | -89 | 90 % |
| | SE 4 | 91.26 ± 0.42 | 735 ± 37 | -92 | -65 | -128 | 0 % |
| | SA 3.1 | 102.37 ± 0.47 | modern | -30 | 4 | -56 | 21 % |
| Sabaju 3 | SA 3.3 | 99.63 ± 0.44 | 30 ± 35 | -39 | 4 | -82 | 3 % |
| | SA 3.6 | 102.63 ± 0.47 | modern | -36 | -14 | -72 | 3 % |





**Figures**

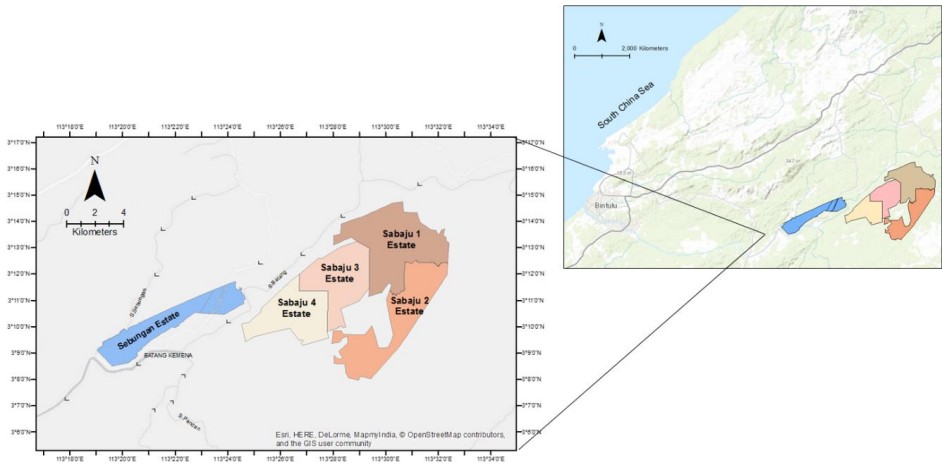

**Fig. 1.** Location of the Sebungan and Sabaju oil palm estates, in Sebauh Bintulu district Sarawak. The estates

are bordered by a network of rivers (left map: grey and white lines; right map: blue lines) namely the Batang

Kemena, Sungai Sebungan, S. Batang and S.Pandan. Arrows indicate direction of water flow.



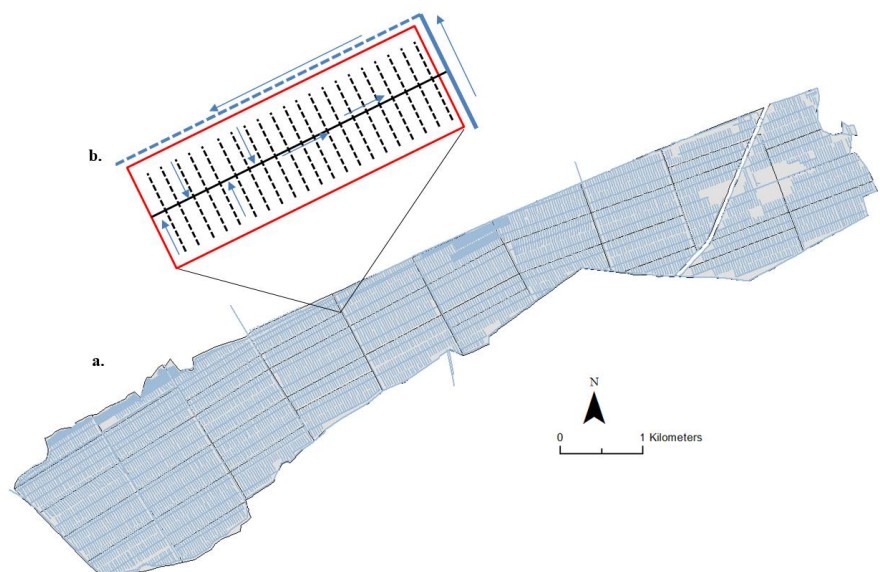

**Fig. 2.** Example of **(a)** peatland oil palm plantation (Sebungan estate) layout with the planting blocks, drains and

roads highlighted and **(b)** close up schematic of typical drainage set up on a peat oil palm planting block. Red

box = planting area, black dashed line = first order field drains, solid black line = second order collection drain,

solid blue line = third order main drain, blue dashed line = perimeter drain. Arrows show the prevailing

direction of water flow



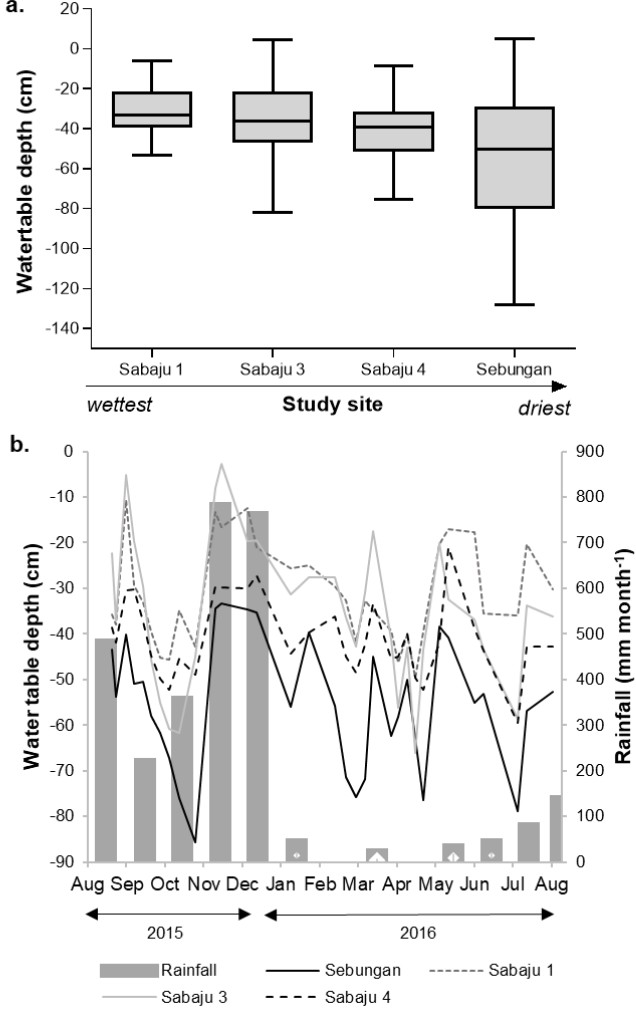

**Fig. 3** Water table depths measured in dip wells in all monitored areas, presented as **(a)** box plots showing mean water table depths for all monitored plantations over the study period (August 2015-September 2016) (central horizontal line), along with the minimum and maximum depths recorded over the entire study period (bars). Box = 75th and 25th percentiles, positioned in sequence from highest to lowest watertable and **(b)** as a time-series of mean weekly water table depths. Negative values indicate that water table was below the peat surface, positive values indicate that there was standing water above the peat surface. Monthly rainfall data were obtained from the rainfall gauge at the Sebungan plantation base (August 2015 –August 2016).





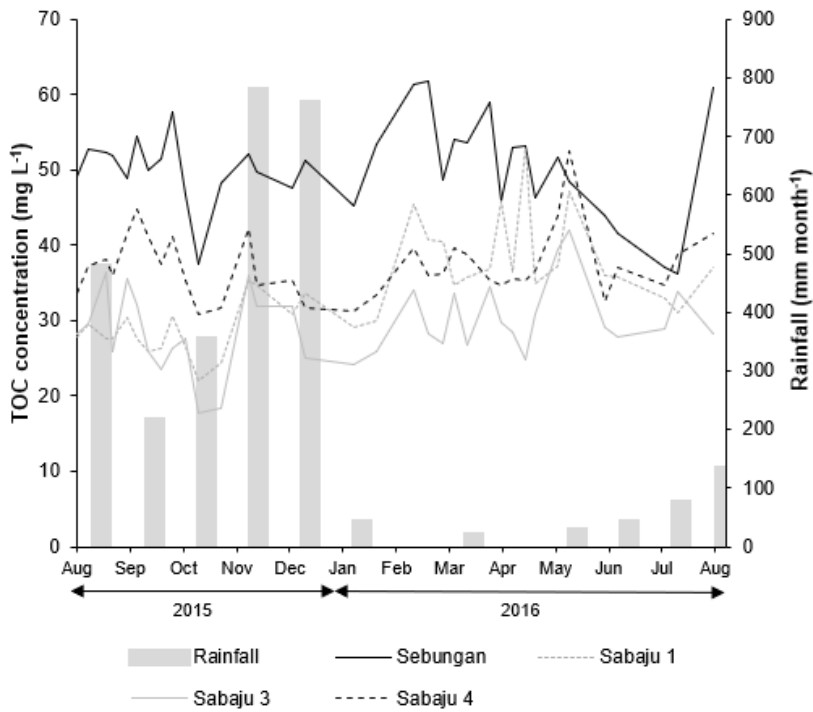

**Fig. 4** Weekly TOC concentration data for the study plantations, alongside monthly rainfall. Data presented are

mean weekly TOC concentrations from all drains within each site. Monthly rainfall data were obtained from the

rain gauge at the Sebungan plantation base (August 2015 –August 2016).




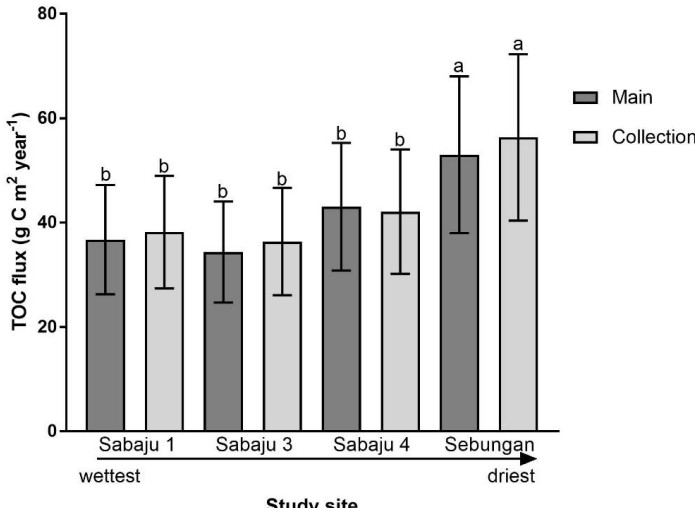

**Fig. 5.** Mean annual TOC fluxes for the plantation drains. Error bars represent ± the 95% confidence interval (CI; standard error x 1.96) which encompasses the propagated error associated with uncertainty in the DOC concentration, discharge and annual runoff derived from the Monte Carlo Simulation. Fluxes are separated into the different drain types; third order (main) and second order (collection). Estates are presented in sequence of drainage intensity. Letters 'a' to 'b' denote significant differences ($p < 0.05$, unpaired, one-way ANOVA) across the study sites irrespective of drain type.



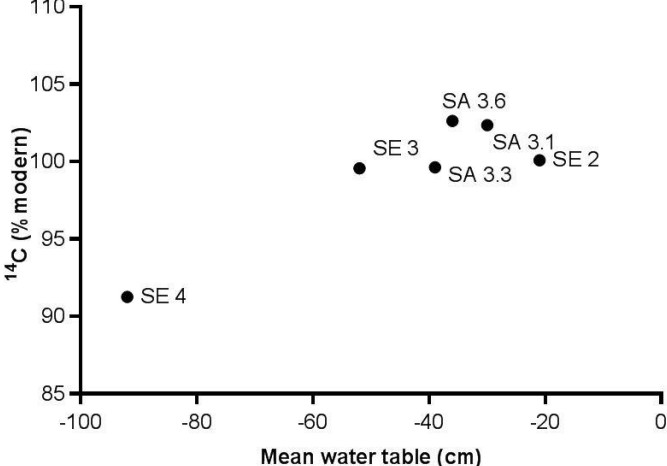

**Fig. 6** Mean water table depths plotted against **(a)** $^{14}$C (% modern) for all six sites. Negative numbers denote

distance below the peat surface



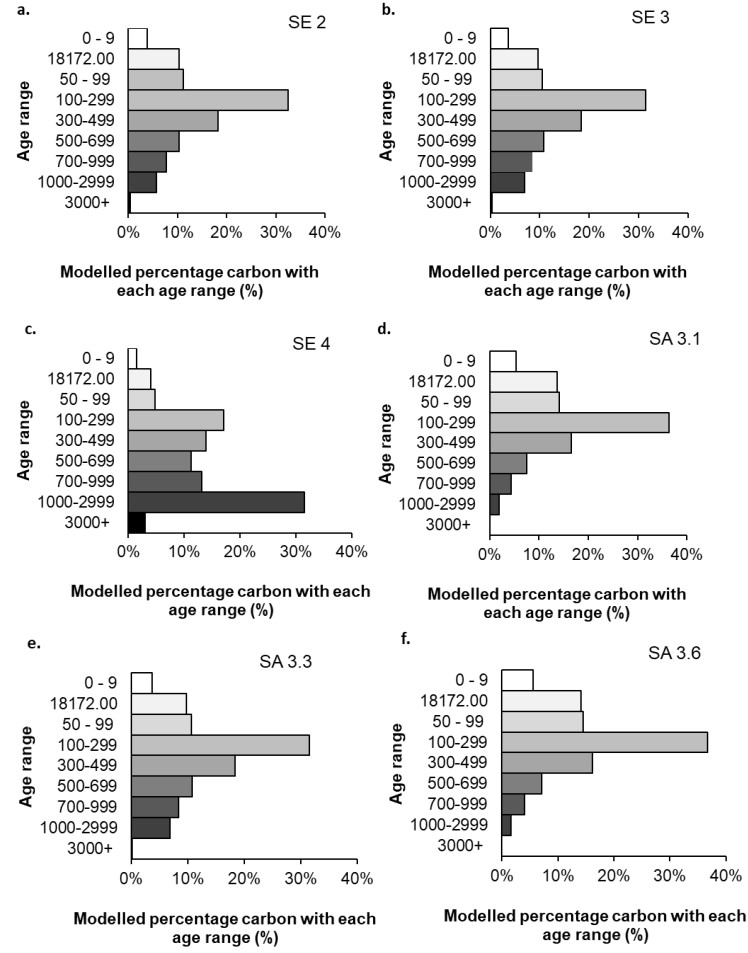

**Fig. 7** Modelled age distributions of $DO^{14}C$, as estimated from the age attribution model for all radiocarbon

dated water samples (n=6).