# Peer review of "Fluvial organic carbon fluxes from oil palm plantations on tropical peatland"

_Biogeosciences, 2018_

## Referee Comment (RC1) · Anonymous Referee #1 · 6 Nov 2018

General comments:

This study provides a quantification of TOC fluxes from an oil palm plantation on tropical peatlands, as well as characterization of aromaticity and 14C content of DOC released from these plantations. Tropical peatlands have been massively converted to plantations. Their contribution to carbon export to surface water and to the ocean has been suggested by the handful studies existing in the area. Although not novel, this study is an important contribution to the literature since it provides quantification of TOC fluxes in the context of oil palm plantations. I therefore recommend publication after minor revisions.

Detailed comments:

[Figure]

The authors measured both DOC and POC. This information is somehow lost as only the TOC fluxes are reported in the abstract and most of the figures. In the abstract, the percentage of DOC should be indicated instead of "TOC fluxes [. . .] were dominated by DOC". The relative proportion of DOC and POC in the TOC flux should also be added in Figure 3.

Abstract:

The values of TOC flux reported in the abstract should be tempered by the fact that the study is based on a single year survey, strongly influenced by an el Ninõ event, and therefore lower discharge than usually observed.

Method:

Section 2.9: Specify the corer type you used.

Results:

Section 3.4 : is the correlation significant?

Discussion:

The discussion section on bulk density and carbon stocks is not clear. Since no carbon content were measured, it is difficult to discuss the carbon density values. The differences are only based on bulk density differences (higher in Sebungan). The link with higher TOC fluxes is not established. This section should be improved or removed from the manuscript.

Figures :

Figure 1 should be improved. A larger map of Borneo with the location of the site would be useful. On the detailed map, Lat/Long are not readable.

Table :

SE3 and SE4: There might be a mistake in the % of time water table was below -60 cm

(0 % for the mean water table of -92 cm, and 90% for the mean water table of -52 cm)

References:

Some references cited in the text do not appear in the reference list (Jones et al., 2016, Gandois et al., 2014).

---

## Author Comment (AC1) · 9 Nov 2018

We thank the reviewer for their constructive comments on our manuscript. Here we give our initial response to these comments, and will provide a modified manuscript after the discussion is closed.

General comments: - the percentage of DOC will be indicated in the abstract. A division will be added to Figure 5 (I think that was the graph the reviewer meant) that separates the TOC flux into the relative DOC and POC contribution. An explanation will be added into the Figure 5 caption.

Abstract: -The observation that this study was carried out on a single year and during an El Ninõ event will be added into Lines 9/10 in the abstract

Methodology: -The corer type will be specified in Lines 20/21

Results:

The correlation significance between watertable and radiocarbon data will be added into Line 29.

Discussion: - an extra sentence will be added into Line 25 to make it clear that as no carbon content was measured the link between peat bulk density and TOC fluxes cannot be fully established.

Figure 1: -a new figure will be added that shows the location of the sites relative to the island of Borneo and clearer Lat/Long coordinates added.

Table 3: Thank you for highlighting the odd '% of water time below -60cm' data mistake. This data will be reanalysed and changed in Table 3.

References: - Gandois et al., 2014 was mistyped and was supposed to be 2013,. Gandois et al., 2014 will removed from the manuscript and replaced with 2013. Jones et al., 2016 will be added to the references list.

---

## Referee Comment (RC2) · Anonymous Referee #2 · 22 Nov 2018

**General comments**

This study evaluates the fluvial carbon fluxes from oil palm plantation on tropical peat. The contribution of fluvial carbon fluxes to carbon balance in tropical peat ecosystem has not been understood yet. Thus, this study provides important and valuable information to stakeholder involvement in this field. Therefore, I recommend publication after several revisions as below.

**Major points**

**1. Validity of the assumption of RE**

The authors assume that meteorology, soil properties, and topology are similar among the plantations because they are located close to each other, and apply the same value

of RE to all the plantations (P9 L15-18). However, bulk density was 1.8 times higher in Sebungan than in Sabaju (Table 1) possibly it is significantly different. The higher bulk density shows lower soil porosity, suggesting that different water storage and RE between SA and SE. Please add more discussion about the validity of the assumption of RE.

2. Temporal variation of discharge

The authors mentioned that the temporal variation of discharge is larger than that of the DOC concentration so that the temporal change in DOC flux is strongly influenced by discharge compared with DOC concentration. I agree with this opinion. However, the author did not show the temporal variations in discharge through the temporal variations in TOC concentration was shown. Thus, I would like to recommend to show the temporal variations of discharge, too.

Minor points

P3 L27-28: I don't know a paper that land compaction by heavy machine increase peat decomposition. But I know the opposite results, for example

* Melling et al. (2005) Soil CO2 flux from three ecosystems in tropical peatland of Sarawak, Malaysia. Tellus, 37B, 1445-1453.

* Othman et al. (2011) Best management practices for oil palm cultivation on peat: Ground water-table maintenance in relation to peat subsidence and estimation of CO2 emissions at Sessang, Sarawak. Journal of Oil Palm Research, 23, 1078-1086.

P14 L1: Is this calculated RE the mean or median of Monte Carlo simulation shown in Fig. S4?

P14 L6: Probably, "and" after 49.6 mg l-1 is not necessary.

Table 3 & Fig. 7: Please explain what is SA 3.1, 3.3, and 3.6.

Fig. 3: It seems that there is no relationship between the rainfall pattern and the temporal variations in water table depth, which is not common, possibly because the rainfall is monthly data whereas water table depth is weekly data. Please check whether there is the relationship between rainfall and water table depth, and improve the figure if the water table depth is influenced by the rainfall. If there is no relationship, please discuss why.

Supporting information: Please explain what is SA 1.4 to SA 4.4

---

## Author Comment (AC2) · 26 Nov 2018

We are grateful to Reviewer 2 for their constructive comments regarding our manuscript. Our initial responses to these comments are included below which will be incorporated into the final manuscript, along with a full overview of all changes made.

Major points

1. Validity of the assumption of RE

Reviewer 2: The authors assume that meteorology, soil properties, and topology are similar among the plantations because they are located close to each other, and apply the same value of RE to all the plantations (P9 L15-18). However, bulk density was 1.8

times higher in Sebungan than in Sabaju (Table 1) possibly it is significantly different. The higher bulk density shows lower soil porosity, suggesting that different water storage and RE between SA and SE. Please add more discussion about the validity of the assumption of RE.

Authors response: The reviewer raises an important point regarding our runoff assumption for all sites. We acknowledge that this is a limitation. However, without additional data (i.e. hydraulic conductivity values) it is hard to assume the water storage potential of these different plantations. Thus, for simplicity we assumed a uniform runoff for all sites and believe that is adequate for addressing our main research aim. An additional sentence will be added into the text to acknowledge the simplicity of this assumption, as suggested by Reviewer 2.

2. Temporal variation of discharge

Reviewer 2: The authors mentioned that the temporal variation of discharge is larger than that of the DOC concentration so that the temporal change in DOC flux is strongly influenced by discharge compared with DOC concentration. I agree with this opinion. However, the author did not show the temporal variations in discharge through the temporal variations in TOC concentration was shown. Thus, I would like to recommend to show the temporal variations of discharge, too.

Authors response: A new figure will be added which will show the mean TOC concentrations across all sites alongside the mean discharge values for the different channels. We hope that this will help the visualisation of the relatively stable TOC concentrations in contrast to fluctuating discharge values.

Minor points

Reviewer 2: P3 L27-28: I don't know a paper that land compaction by heavy machine increase peat decomposition. But I know the opposite results, for example * Melling et al. (2005) Soil $CO_2$ flux from three ecosystems in tropical peatland of Sarawak,

Malaysia. Tellus, 37B, 1445-1453. * Othman et al. (2011) Best management practices for oil palm cultivation on peat: Ground water-table maintenance in relation to peat subsidence and estimation of CO2 emissions at Sessang, Sarawak. Journal of Oil Palm Research, 23, 1078-1086.

Authors response: A recent paper by Tonks et al. (2017) closely links the degree of decomposition to the physical properties of peat namely bulk density, shear strength and porosity. This reference will be added to the manuscript to reinforce the points raised by Reviewer 2.

Reviewer 2: P14 L1: Is this calculated RE the mean or median of Monte Carlo simulation shown in Fig. S4?

Author response: This is the mean calculated RE, and will be specified in the text.

Reviewer 2: P14 L6: Probably, "and" after 49.6 mg l-1 is not necessary.

Authors response: This will be omitted.

Reviewer 2: Table 3 & Fig. 7: Please explain what is SA 3.1, 3.3, and 3.6.

Authors response: These are individual sample sites within the Sabaju 3 plantation. This will be clarified in Table 3 and Fig. 7 captions.

Reveiwer 2: Fig. 3: It seems that there is no relationship between the rainfall pattern and the temporal variations in water table depth, which is not common, possibly because the rainfall is monthly data whereas water table depth is weekly data. Please check whether there is the relationship between rainfall and water table depth, and improve the figure if the water table depth is influenced by the rainfall. If there is no relationship, please discuss why.

Authors response: The correlation between these two variables will be checked and will be subsequently discussed within the manuscript.

Reviewer 2: Supporting information: Please explain what is SA 1.4 to SA 4.4

Author response: A sentence will be added to explain what these codes relate to within the Supplementary Material.

---

## Author Response (AR1)

**Authors' Response**

We are grateful to both of the reviewers and the associate editor for their constructive comments on our manuscript. We have revised the manuscript according to the points raised by both reviewers, and have modified the text and figures accordingly. Our responses to each reviewer's comment are listed below in blue italic font.

**Reviewer one comments:**

General:

The authors measured both DOC and POC. This information is somehow lost as only the TOC fluxes are reported in the abstract and most of the figures. In the abstract, the percentage of DOC should be indicated instead of "TOC fluxes [*: : :*] were dominated by DOC". The relative proportion of DOC and POC in the TOC flux should also be added in Figure 3

*The percentage of DOC is now indicated in the abstract (Line 7: "(TOC) fluxes from the plantation second and third order drains were dominated (91%) by dissolved organic"*

*A division has been added to Figure 5 (I think that was the graph the reviewer meant, now Figure 6) that separates the TOC flux into the relative DOC and POC contribution. An explanation has been added into the Figure 6 caption: "Horizontal bar lines represent contribution of DOC (bottom segment) and POC (top segment) to the overall TOC flux".*

Abstract:

The values of TOC flux reported in the abstract should be tempered by the fact that the study is based on a single year survey, strongly influenced by an el Ninõ event, and therefore lower discharge than usually observed.

*The observation that this study was carried out on a single year and during an El Ninõ event has been added in to Lines 9/10 in the abstract: "These fluxes represent a single year survey which was strongly influenced by an El Ninõ event and therefore lower discharge than usual was observed."*

Methodology:

Section 2.9: Specify the corer type you used.

*The corer type has now been specified in Page 12 Line 25*

Results:

Section 3.4 : is the correlation significant?

*The correlation significance between watertable and radiocarbon data has been added into Page 15 Line 7: "Conventional mean DO$^{14}$C age was positively correlated (p < 0.05)…"*

Discussion:

The discussion section on bulk density and carbon stocks is not clear. Since no car- bon content were measured, it is difficult to discuss the carbon density values. The differences

are only based on bulk density differences (higher in Sebungan). The link with higher TOC fluxes is not established. This section should be improved or removed from the manuscript.

*An extra sentence has been added into Page 17 Lines 4-6 to make it clear that as no carbon content was measured the link between peat bulk density and TOC fluxes cannot be fully established: "However, as peat carbon content was not measured the link between peat bulk density and the TOC fluxes cannot be fully established."*

Figure 1:

Figure 1 should be improved. A larger map of Borneo with the location of the site would be useful. On the detailed map, Lat/Long are not readable.

*A new figure has been added that shows the location of the sites relative to the island of Borneo and clearer Lat/Long coordinates added.*

Table 3:

SE3 and SE4: There might be a mistake in the % of time water table was below -60 cm  (0 % for the mean water table of -92 cm, and 90% for the mean water table of -52 cm)

*Thank you for highlighting the odd '% of water time below -60cm' data mistake. This data has been reanalysed and changed in Table 3.*

References:

Some references cited in the text do not appear in the reference list (Jones et al., 2016, Gandois et al., 2014).

*Gandois et al., 2014 was mistyped and was supposed to be 2013,. Gandois et al., 2014 has been removed from the manuscript and replaced with 2013.  Jones et al., 2016 has been added to the references list.*

**Reviewer two comments:**

Major points

1. Validity of the assumption of RE

The authors assume that meteorology, soil properties, and topology are similar among the plantations because they are located close to each other, and apply the same value of RE to all the plantations (P9 L15-18). However, bulk density was 1.8 times higher in Sebungan than in Sabaju (Table 1) possibly it is significantly different. The higher bulk density shows lower soil porosity, suggesting that different water storage and RE between SA and SE. Please add more discussion about the validity of the assumption of RE.

*The reviewer raises an important point regarding our runoff assumption for all sites. We acknowledge that this is a limitation. However, without additional data (i.e. hydraulic conductivity values) it is hard to assume the water storage potential of these different plantations. Thus, for simplicity we assumed a uniform runoff for all sites and believe that is*

*adequate for addressing our main research aim. An additional sentence has been added into the text (Page 9, Lines 18-21) to acknowledge the simplicity of this assumption: "To do this, $R_E$ was assumed to be the same for all plantation sites, based on the assumption that all sites were hydrologically similar in terms of the annual water balance. While this is a simplistic approach all sites had similar soil properties, topography, vegetation and management and were sufficiently close together such that they experienced very similar rainfall".*

2. Temporal variation of discharge

The authors mentioned that the temporal variation of discharge is larger than that of the DOC concentration so that the temporal change in DOC flux is strongly influenced by discharge compared with DOC concentration. I agree with this opinion. However, the author did not show the temporal variations in discharge through the temporal variations in TOC concentration was shown. Thus, I would like to recommend to show the temporal variations of discharge, too.

*A new figure has been added (Fig. 5) which shows the mean TOC concentrations across all sites alongside the mean discharge values for the different channels. This helps to highlight the relatively stable TOC concentrations in contrast to fluctuating discharge values. This figure is subsequently referenced on Page 14 Lines 19/20 Page 15 Line28/29.*

Minor points

P3 L27-28: I don't know a paper that land compaction by heavy machine increase peat decomposition. But I know the opposite results, for example

* Melling et al. (2005) Soil CO2 flux from three ecosystems in tropical peatland of Sarawak, Malaysia. Tellus, 37B, 1445-1453.

* Othman et al. (2011) Best management practices for oil palm cultivation on peat: Ground water-table maintenance in relation to peat subsidence and estimation of CO2 emissions at Sessang, Sarawak. Journal of Oil Palm Research, 23, 1078-1086.

*A recent paper by Tonks et al. (2017) closely links the degree of decomposition to the physical properties of peat namely bulk density, shear strength and porosity. This reference has been added to Pg 3 Line 28 and to the references list (Page 23 Line 6).*

*In the studies by both Melling et al. (2005) and Othman et al. (2011) the autotrophic (tree roots) and heterotrophic (peat oxidation) respiration emissions are not separated. As such, it is hard to draw conclusions about the total net ecosystem-atmosphere $CO_2$ exchange. The implications of this data are elaborated by Page et al. (2011; Review of peat surface greenhouse gas emissions from oil palm plantations in Southeast Asia. White Paper Number 15).*

P14 L1: Is this calculated RE the mean or median of Monte Carlo simulation shown in Fig. S4?

*This is the mean calculated RE, this has now been specified on Page 14 Line 5*

P14 L6: Probably, "and" after 49.6 mg l-1 is not necessary.

*This has been omitted Page 14 Line 13.*

Table 3 & Fig. 7: Please explain what is SA 3.1, 3.3, and 3.6.

*These are individua sample sites within the Sabaju 3 plantation. This has hopefully now been clarified within the Table 3, Figure 7 and Figure 8 captions.*

Fig. 3: It seems that there is no relationship between the rainfall pattern and the temporal variations in water table depth, which is not common, possibly because the rainfall is monthly data whereas water table depth is weekly data. Please check whether there is the relationship between rainfall and water table depth, and improve the figure if the water table depth is influenced by the rainfall. If there is no relationship, please discuss why.

*The lack of correlation is now explained on Page 13 Lines 21 – 24: "relationship between the rainfall pattern and temporal variability in the water table depth could not be drawn due to differences in the data resolution (i.e. monthly data = rainfall; weekly data = water table depth)".*

Supporting information: Please explain what is SA 1.4 to SA 4.4

*A sentence explaining what these codes represent has been added as an end sentence to the introductory paragraph of the supplementary material: "Individual monitoring sites across the plantation estates are quoted as follows: Sebungan (SE 1, SE 2, SE 3, SE 4); Sabaju 1 (SA 1.1, SA 1.2, SA 1.3, SA 1.4), Sabaju 3 (SA 3.1, SA 3.5, SA 3.6); Sabaju 4 (SA 4.1, SA 4.2, SA 4.3, SA 4.4)"*